# Alzheimer’s Disease Treatment: The Search for a Breakthrough

**DOI:** 10.3390/medicina59061084

**Published:** 2023-06-04

**Authors:** Allison B. Reiss, Dalia Muhieddine, Berlin Jacob, Michael Mesbah, Aaron Pinkhasov, Irving H. Gomolin, Mark M. Stecker, Thomas Wisniewski, Joshua De Leon

**Affiliations:** 1Department of Medicine and Biomedical Research Institute, NYU Long Island School of Medicine, Mineola, NY 11501, USA; dalia.mddn@gmail.com (D.M.); bjacob05@nyit.edu (B.J.); mmesbah26@gmail.com (M.M.); aron.pinkhasov@nyulangone.org (A.P.); irving.gomolin@nyulangone.org (I.H.G.); joshua.deleon@nyulangone.org (J.D.L.); 2Fresno Institute of Neuroscience, Fresno, CA 93730, USA; mmstecker@gmail.com; 3Center for Cognitive Neurology, Departments of Neurology, Pathology and Psychiatry, NYU School of Medicine, New York, NY 10016, USA; thomas.wisniewski@nyulangone.org

**Keywords:** Alzheimer’s disease, amyloid, inflammation, dementia, drug therapy, diet, lifestyle

## Abstract

As the search for modalities to cure Alzheimer’s disease (AD) has made slow progress, research has now turned to innovative pathways involving neural and peripheral inflammation and neuro-regeneration. Widely used AD treatments provide only symptomatic relief without changing the disease course. The recently FDA-approved anti-amyloid drugs, aducanumab and lecanemab, have demonstrated unclear real-world efficacy with a substantial side effect profile. Interest is growing in targeting the early stages of AD before irreversible pathologic changes so that cognitive function and neuronal viability can be preserved. Neuroinflammation is a fundamental feature of AD that involves complex relationships among cerebral immune cells and pro-inflammatory cytokines, which could be altered pharmacologically by AD therapy. Here, we provide an overview of the manipulations attempted in pre-clinical experiments. These include inhibition of microglial receptors, attenuation of inflammation and enhancement of toxin-clearing autophagy. In addition, modulation of the microbiome-brain-gut axis, dietary changes, and increased mental and physical exercise are under evaluation as ways to optimize brain health. As the scientific and medical communities work together, new solutions may be on the horizon to slow or halt AD progression.

## 1. Introduction

Alzheimer’s disease (AD) is a progressive, fatal neurodegenerative condition that presents clinically as impairment of cognitive function and memory along with changes in behavior and personality [1,2]. Neuronal loss and synaptic dysfunction are hallmarks of the disease. Detected microscopically within the brain are amyloid plaques formed by aggregation of amyloid β and neurofibrillary tangles composed of hyperphosphorylated tau protein [3,4]. Increasing global concern has led to the allocation of extensive resources to study AD pathophysiology, but our understanding of its causes remains rudimentary, and our treatments are inadequate [5,6].

Currently, fully approved AD treatments are limited to acetylcholinesterase inhibitors and N-methyl d-aspartate receptor antagonists. These agents address some AD symptoms but are not disease-modifying [7,8]. Recently, the FDA partially approved the anti-amyloid human immunoglobulin (Ig)G1 monoclonal antibodies aducanumab and lecanemab [9,10,11,12]. Aducanumab, the first new therapy for AD since 2003, was approved by the FDA via an accelerated approval process. The effectiveness of this drug has been called into question, particularly since the FDA’s own Advisory Committee voted against its release [10,13]. It carries serious risks of amyloid-related imaging abnormalities (ARIA)—edema or hemorrhage [14,15]. Lecanemab in Phase III testing showed more clear cognitive benefits, slowing cognitive decline by 27% on the Clinical Dementia Rating-Sum of Boxes (CDR-SB) scale at 18 months [11,12]. These relatively modest clinical benefits are also associated with the potential for significant ARIA complications. Other drugs with a similar mode of action are in development [16,17]. However, the impact of this drug class on AD is not curative and, at best, may modestly slow progression. The need for a more significant leap forward remains.

This review will survey the newest approaches to AD therapy beyond amyloid and tau, hoping that one or more of these may lead to true advances in conquering this devastating disease.

## 2. Finding a Viable Approach

Studies in humans indicate that eliminating or clearing amyloid-β (Aβ) or tau does not halt or reverse AD [18,19,20]. This calls into question the assertion that the Aβ oligomer is the primary initiator of AD. Instead, Aβ and tau protein likely appear after the damage is too extensive for repair, or they are indications of a pathological process and not the cause.

The multifactorial etiology of AD likely involves impaired regulation of multiple signaling pathways, ultimately leading to neuronal and synaptic loss and hypoplasticity [21,22]. AD neuronal death can be attributed to mitochondrial dysfunction, DNA oxidative damage, chronic neuroinflammation and failure of cellular repair mechanisms [23,24]. Ultimately, the preservation of neuronal function and prevention of neuronal loss is the goal of any cognition-preserving AD treatment.

## 3. Inflammation in AD

### 3.1. Overview

Aβ plaques and neurofibrillary tangles of tau protein are hallmarks of AD and indicators of neurological pathology that manifest years or decades before an official AD diagnosis [25,26]. However, therapies directed at these deposits have not shown therapeutic results in humans, and only a few symptomatic treatments for some patients with AD are currently available [27,28,29]. There is no cure, but studies over the years have shown that there may be causative agents that act via the promotion of neuroinflammation, which may lead to Aβ and tau accumulation as well as neuronal destruction [30]. In the following subsections, we discuss several anti-inflammatory drugs being considered for repurposing in treating AD and newly developed agents that can interfere with destructive inflammatory pathways in the neuron (Table 1).

### 3.2. Neuroinflammation and Microglia

Neuroinflammation can be defined as a sustained immune response in the CNS. Acute inflammation can help defend against insults to the brain, such as toxins, infection, or injury [31,32]. However, in the chronic phase, there can be a cycle of increased inflammation and further damage due to excessive activation of immune cells such as microglia, which can migrate and release proinflammatory cytokines [33]. Historically, immune antigens found around amyloid plaques in AD have been reported in studies since the 1980s. The findings of cytokines and activated complement factors were reported in the 1990s. This opened the door to the hypothesis that immunological processes are involved in the pathology of degenerative CNS diseases such as AD, schizophrenia, and Parkinson’s disease [34,35].

In AD, microglia and astrocytes are the resident immune cells activated in the parts of the brain affected by Aβ plaques and tau NFTs [36]. Microglia are cells of mesodermal origin, and the most abundant immune cells present in the brain. Normally in the resting state of a healthy brain, they maintain homeostasis of the neuronal environment, control the proliferation and differentiation of neurons, and perform immune surveillance [37,38]. However, Microglia are dynamic, even in the resting state, constantly moving their fine cellular processes to execute their functions of phagocytosing cellular debris and regulating neural plasticity and synaptic formation [39,40].

When microglia detect injury or disease to the CNS, they become activated and change from ramified to amoeboid morphology and a pro-inflammatory phenotype [41]. They change appearance through cellular enlargement and retraction of their processes. In addition to the physical changes, microglia mount a host defence by releasing inflammatory mediators such as cytokines, chemokines, free radicals, and reactive oxygen species, which, in cases of overactivation, can be toxic to the brain [42]. When not over-exuberant, microglia have been shown to gather pathological debris and have positive effects as they clear Aβ plaques, as demonstrated in multiple animal model systems [43]. They release both neurochemicals with neuroprotective effects and neurotoxic mediators [44]. Constantly activated microglia, over prolonged periods, will become less able to clear Aβ plaques and peripheral macrophages are then activated, which further exacerbate amyloid and tau pathology as they surround the damaged areas. In the process, pro-inflammatory products are additionally released, and oxidative damage ensues, creating a cycle of damage [45]. It has even been shown that the release of cytokines such as IL-1 exacerbates amyloid pathology while IL-6 stimulates the kinase CDK5, which is a main mechanism in the tau hyperphosphorylation mechanism [46,47]. These findings have inspired the idea that inflammation may be the link between these two novel pathways.

Traditionally, microglia have been categorized into classical (M1) and alternative (M2) phenotypes, with a range of intermediate phenotypes occurring [48]. M1 microglia release inflammatory mediators, produce ROS, and contribute to neuronal damage, whilst M2 microglia release anti-inflammatory mediators, promote inflammation resolution, and are neuroprotective [49]. These two opposing types play a role in neurodegenerative diseases, including AD, multiple sclerosis and Parkinson’s disease and have led to the study of balancing M1 and M2 polarization for increasing neuroprotection [44,50]. Although the canonical M1/M2 paradigm may be helpful, it should be noted that refinements in defining microglial state can yield a more accurate profile, and transcriptomics are applied to account for subtleties in phenotype in normal and AD cells [51].

### 3.3. Anti-Inflammatory Drug Repurposing as an Approach to AD via Microglia

M1 inhibitive agents such as non-steroidal anti-inflammatory drugs (NSAIDs), which act by inhibiting cyclooxygenases (COX) 1 and 2, enzymes that catalyze the conversion of arachidonic acid to prostaglandins, have not shown benefit in treating AD [52]. COX-2 is over-expressed in activated microglia, and thus it was reasoned that COX-2 inhibition might reduce neuroinflammatory activity and prostaglandin release by these cells [53]. Initially, throughout the late 20th century, several case-control retrospective epidemiological studies showed that rheumatoid arthritis patients who were on chronic NSAIDS had decreased severity and progression of AD as compared to non-NSAID users [54,55]. However, human trials showed variable outcomes with no positive conclusion. A meta-analysis of seven studies which included the NSAIDs diclofenac/misoprostol, nimesulide, naproxen, rofecoxib, ibuprofen, indomethacin, tarenflurbil, and celecoxib, showed the clinical significance of NSAIDs treatment compared with placebo when patients were assessed by cognitive and memory exams. However, studies were limited by study size [56]. This discrepancy between epidemiological and prior research studies has partly been attributed to the time NSAIDs need to provide a protective and/or therapeutic effect. This hypothesis was explored by the Baltimore Longitudinal Study of Aging, which showed that the risk of AD was reduced after two years of NSAID use. However, no conclusions could be made on protective benefit in terms of cognitive decline or the specific NSAID that conferred the most benefit. In addition, long-term NSAID use is associated with risks of gastric ulceration, bleeding, and nephrotoxicity, which may not be suitable for many patients depending on their medical conditions [57]. The more recent INTREPAD study observed the effects of naproxen in people who had a strong family history of AD but without an official diagnosis. One hundred people were prescribed naproxen, and the remaining 100 a placebo, and the new Alzheimer Progression Score (APS) was used to predict the onset of the clinical disease over the coming decade or more. The results proved negative, with no evidence that the APS was reduced with naproxen [58].

Recent work also shows that more modern disease-modifying anti-rheumatic agents with anti-inflammatory properties do not reduce AD risk [59].

### 3.4. Repurposing Anti-TNF Agents

Pro-inflammatory markers released by activated microglia, such as tumor necrosis factor (TNF)-α, have also been used as a target for AD therapies [60,61]. TNF-α can interact with the 55-kDa TNF receptor 1 (TNFR1) to induce a neuroinflammatory state, or it can interact with the 75-kDa TNF receptor 2 (TNFR2) to produce a neuroprotective effect [62]. Given this duality, therapies currently underway include TNF-α blockade, inhibition of TNFR1 signaling or induction of TNFR2 signaling. Etanercept, an anti-TNF-α antibody that is a fusion protein between a human IgG1 Fc-tail and TNFR2, has been studied in murine models of AD with Aβ plaque formation and found to decrease TNF-α levels, reduce neuronal injury and improve cognitive measures [63,64]. In addition, intra-cerebral administration of the chimeric anti-TNF-α antibody infliximab to mice overexpressing APP reduced the formation of both Aβ- plaques and tau neurofibrillary tangle [65].

A second-generation biologic TNF-α inhibitor, XPro1595, is a PEG-ylated mutant form of TNF that complexes with TNF-α in a way that prevents it from binding to TNFR1 [66]. XPro1595 has been studied pre-clinically in AD mice and human clinical trials. For example, the XPro1595 treatment of 5XFAD Aβ-overexpressing mice decreased Aβ plaques and reduced immune cell activation [67]. XPro1595 clinical trials have also shown positive results regarding targeting inflammation. For example, a 12-week, phase 1b study, which included weekly injections of 0.03, 1.0 or 3.0 mg/kg XPro1595 in mild-to-moderate AD patients, showed a 40.6% reduction in arcuate fasciculus inflammation, an area of the brain responsible for intra-cerebral connections, short term memory and language [68].

### 3.5. Inciting the M2/TREM 2 Phenotype in Microglia

Another pathway researchers have taken is to study the activation of M2 microglia to amplify the neuroprotective effects. Genetic mutations in microglial and cytokine receptors also corroborate the neuroinflammatory link to AD [69]. The most significant lead in recent studies has found that heterozygous mutations in the M2 microglia regulator known as Triggering Receptor Expressed on Myeloid Cells 2 (TREM2) increased the risk of AD significantly. Initially, TREM2 was studied after gene sequencing revealed that this receptor’s homozygous loss of function mutation led to an autosomal recessive disease known as Nasu-Hakola disease, which involves early-onset dementia and bone pathology [70]. Given its link to progressive dementia, a study was conducted using genome, exome, and Sanger sequencing to analyze the genetic variability in TREM2 in 1092 patients with AD and 1107 controls. Results showed more variants on exon 2 of the TREM2 gene in AD patients, with rs75932628 (encoding R47H) found to be the most common variant. This R47H mutation showed a highly significant association with AD (*p* < 0.001) [71]. An agonist TREM2 mouse IgG1 antibody (AL002a) developed to activate TREM2 signaling in vivo was administered intracranially to 5XFAD Aβ-overexpressing mice. The AL002a caused activation and recruitment of microglia to amyloid plaques, decreased Aβ deposition and improved memory and cognition in these mice [72].

Similarly, AL002c, a mouse IgG1 anti-human TREM2 monoclonal antibody, was studied in 5XFAD mice carrying the common variant (CV) of TREM2 and in 5XFAD mice carrying the R47H loss-of-function Trem2 mutation. An injection of AL002a increased the phagocytic activity of the microglia and reduced Aβ plaque toxicity in both types of mice [73]. In addition, a Phase I clinical trial of AL002 (NCT03635047) found the antibody to be safe and tolerable in healthy adults with mild-to-moderate AD, and the levels of TREM2 in CSF were found to be decreased in a dose-dependent fashion after a single intravenous injection of AL002. These favorable results have led to a currently ongoing Phase 2 randomized, double-blind, placebo-controlled clinical trial which examines the role of AL002 use in patients diagnosed with the early stages of AD [74].

### 3.6. CD33

CD33, a member of the family of sialic acid-binding immunoglobulin-like lectins, is a transmembrane receptor expressed on microglia that affects microglial phagocytosis [75]. Genome-wide association studies have revealed an association between late-onset AD and polymorphisms in CD33 [76,77].

In the AD brain, CD33 levels and the number of CD33+ microglia are increased, and higher CD33 expression correlates positively with higher Aβ plaque load [78]. In CD33 knockout mice, Aβ plaque burden is reduced. In cell culture studies using the THP-1 human macrophage cell line, knockout of CD33 increased phagocytosis of aggregated Aβ but also increased the inflammatory phagocytic oxidative burst [79]. The AL003 antibody, which binds to CD33, was evaluated in a clinical trial, but although target engagement was confirmed, the antibody is no longer in the pipeline [80,81,82]. The future of CD33 targeting AD remains uncertain, but small molecule binding to CD33 may be an avenue of study [83].

### 3.7. PTI-125

PTI-125, a small molecule AD treatment, binds to an abnormal conformation of filamin A that is induced by Aβ42 and restores the conformation to its native state [84]. In humans, a Phase 2a safety, pharmacokinetics, and biomarker study in 13 AD patients showed that after 28 days of twice daily oral treatment, all patients had a biomarker response to the drug (CSF P-tau decreased 34%, *p* < 0.0001), which was well tolerated, with no drug-related adverse events [85]. However, there is controversy surrounding this drug. While studies are continuing, including an open-label extension study for long-term safety and tolerability, the issue of possible irregularities is not resolved [86].

### 3.8. Role of Peripheral Inflammation in AD

An integrative perspective in relation to AD pathogenesis, specifically exploring systemic metabolic factors such as diabetes and abnormalities in the gut microbiome, has been gaining attention and has raised important questions. One of the first epidemiological studies to demonstrate the association between type 2 diabetes (T2DM) and dementia was the Rotterdam Study. This population-based prospective cohort study started in 1990 and included diabetes as one of the multiple modifiable cardiovascular risk factors. Over 8000 participants were followed over decades, and it was found that in relation to dementia, T2DM had the second most population-attributable risk. This value measures the magnitude of the potential to prevent disease [87]. Other studies have solidified this relationship and shown that glucose utilization is reduced in the AD brain with hypometabolism in specific brain areas on fluorine 18 fluorodeoxyglucose positron emission tomography neuroimaging [88,89,90,91]. Multiple research reports have gone a step further by labeling AD as type 3 diabetes in which insulin resistance can occur systemically, including in the brain and lead to multiple, thus-far unidentified pathways of neurodegeneration [92,93]. It has been postulated that the low-grade inflammatory state seen in persons with T2DM leads to immune activation that affects the brain [94,95,96]. In diabetic rodent models, pro-inflammatory markers, such as IL-2, IL-6 and TNF-α, are increased in the brain [97,98,99].

T2DM can impair autophagy, a vital process needed for clearing toxic reactive oxygen species and other waste, and this may interfere with the clearance of both Aβ and tau [100,101,102]. T2DM is a metabolic disease characterized by dysfunctional insulin secretion and the development of insulin resistance. Insulin affects not only glucose levels in the blood but also neurogenesis and energy metabolism in the brain. It is postulated that diabetes-induced peripheral insulin resistance can promote central insulin resistance [103]. This possibility has prompted the development of brain-available forms of insulin as potential AD treatment. Insulin, with a molecular weight of 5808 Da, is too large to passively cross the (blood-brain barrier) BBB, which limits permeability to 400 Da or less. Thus, extra-neuronal forms have been studied, specifically intranasal insulin. This insulin has been shown to evade the BBB and reach the CNS within 1 h of usage via multiple mouse and human in vitro studies.

Furthermore, its safety profile is low risk because there is minimal systemic absorption and subsequent effects on cortisol and growth hormone if maintained underdosing 200 IU [104,105,106]. The positive impact of intranasal insulin was initially explored in individuals without cognitive impairment. An eight-week trial of 160 IU of intranasal insulin in 38 healthy young male and female participants versus placebo showed improved hippocampal declarative memory via delayed word recall testing. Immediate recall memory testing showed no improvement [107]. Several pilot studies have been performed in men and women with mild to moderate cognitive impairment in which insulin or a placebo was given [108,109]. Memory scores improved, cognitive ability was maintained, and brain volume of the parietal and hippocampal areas was preserved over four months with the treatment. A study looking at intranasal insulin in mild cognitive impairment (MCI) and early AD found that the apolipoprotein (apo)E genotype affected the results such that benefits were greater in those not carrying the apoE4 allele, a known risk factor for AD [110]. The administration of intranasal insulin, although not a cure, may benefit some MCI and AD patients, but more extensive studies of efficacy and mechanism are needed [111,112].

Metformin, which easily penetrates the BBB, is a hypoglycemic drug with neuroprotective properties in animal models [113]. In rats, it protects against an amyloid-induced decline in cognitive function by reducing oxidative stress and neuroinflammatory processes [114]. In addition, Metformin has favorable effects on insulin pathways, and it has shown some promise in human studies [115,116].

The gut has also been explored as a potential link to the progression of inflammation in the brain leading to AD. There is a relationship between the brain and gut, known as the “microbiome-gut-brain axis,” in which the bacterial communities in the gut communicate with the CNS via molecules that act both directly and indirectly to influence behavior (Figure 1) [117]. Communication is bidirectional; thus, the brain can also affect the gut by changing appetite and eating patterns. The gut microbiome consists of many bacterial species residing in the small and large intestines, engaged in a symbiotic relationship with the human body [118]. The gut microbiome is involved in the immune response of the intestines, protecting the host from detrimental bacterial overgrowth and carcinogens by releasing short-chain fatty acid metabolites. Common gut species such as Saccharomyces, Bacillus and Bifidobacterium have been shown to break down short-chain fatty acids and affect the synthesis of dopamine, acetylcholine, glutamate, and serotonin [119,120,121]. These neurotransmitters and signaling molecules produced by bacteria in the gut enter the bloodstream through the enterohepatic circulation and can penetrate the BBB resulting in beneficial or detrimental effects on neuronal health [122].

An early study demonstrating a link between the gut microbiome and the brain was performed in germ-free mice characterized by a complete lack of exposure to microorganisms. These germ-free mice were found to have an amplified response to stress restored via recolonizing the mice with the gut microbiome species Bifidobacterium infantis [123]. They also showed a reduced brain-derived neurotrophic factor (BDNF) level in the cortex and hippocampus. Further, the transplantation of microbiota from mice exposed to chronic unpredictable stress into recipient mice not exposed to stress resulted in anxiety and depression-like behavior in the recipient mice [124]. In accordance with this outcome, when fecal matter from healthy mice was transferred into mice with Parkinson’s disease-like syndrome, this afforded neuroprotection, especially against neuroinflammation [125]. Germ-free mice colonized with gut microbiota from human patients with multiple sclerosis exhibit multiple sclerosis-like autoimmune responses [126]. Fecal microbiota transplantation from an AD mouse model into wild-type mice resulted in memory dysfunction, reduced hippocampal neurogenesis, and increased hippocampal neuroinflammation in the recipients [127]. These and many more studies have corroborated a connection between the brain and the gut.

Negative alteration of the gut microbiome, or dysbiosis, is seen in humans with AD, with a decrease in microbial diversity and, in some reports, an increase in Bacteroidetes species [128,129,130]. Bacteroidetes is an umbrella phylum of many different types of gram-negative bacteria found to incite a pro-inflammatory response from the gut, largely attributable to their outer membrane constituent lipopolysaccharide (LPS), a bacterial endotoxin [131]. Bacteroidetes species have been detected in high levels in Type II DM and Parkinson’s patients [132]. Similarly, postmortem brain tissue from patients with AD found LPS and gram-negative bacterial DNA segments localized around amyloid plaques, which may indicate a link between the bacterial pro-inflammatory response and AD pathology [133].

In contrast, there are gut bacteria that may be beneficial to the CNS. The Bifidobacterium genus, gram-positive bacteria found widely in the gastrointestinal tract, have anti-inflammatory effects, and are used in probiotic products [134,135]. Murine studies using cognitively impaired mice injected with LPS showed that administering Bifidobacterium by oral gavage decreased LPS levels and improved cognitive function [136,137]. In human AD studies, which have been limited and with a small population size, there have also been some promising results. A double-blind, controlled clinical trial consisting of 30 AD patients randomized into a group of 30 taking a mix of probiotics (including Bifidobacterium) in milk and a group of 30 consuming milk without added probiotics showed a statistically significant improvement in mini-mental status exam scores in the group taking probiotics after 12 weeks [138]. Studies investigating the microbiome’s association with AD are ongoing with the hope that specific strains of bacteria or combinations of strains may serve as a preventative measure in the clinical course of AD [139].

## 4. Delivery Systems to the Brain Crossing the BBB

Reaching the brain regions affected by AD is challenging, especially because the BBB, through low permeability and active efflux, blocks penetration into the CNS of many drugs and compounds [140]. Therefore, avoiding direct and invasive access to the CNS via methods such as intrathecal or intracerebroventricular injection is a high priority. Instead, it may be possible to use the circulatory system or the nose-to-brain route [141,142]. Lipophilic nanoparticles and biocompatible nanogels composed of hydrophilic polymers are a few technologies for delivery to the brain parenchyma [143]. Targeting the brain reduces the dosage needed and any accompanying toxicities by narrowing the distribution of the medication. In addition, encapsulation can prevent rapid metabolism and elimination and binding to plasma proteins [144].

Nanoparticles range in size from approximately 10 to 100 nm and can be organic or inorganic (often silicon or metallic) [145]. Organic nanoparticles consist of biomaterials such as liposomes, micelles, or polymers (natural or synthetic) that hold the pharmaceutical agent and can penetrate the BBB for site-targeted delivery in the case of the CNS. Designing a coated nanoparticle is a strategy that combines many advantages in traversing the BBB with minimal toxicity and immunogenicity, and good targeting. The technique involves coating the nanoparticle with a cell membrane-like phospholipid bilayer outer covering over a lipid-based or polymeric core that holds the drug [146,147]. Conjugation of ligands onto the nanoparticle surface can bring customized ligand-receptor binding and internalization of the particle in the desired cell type via receptor-mediated endocytosis [148]. Nanoparticles can also be used to carry oligonucleotides to employ antisense technology to alter gene expression [149].

Nanoparticles are a potential new tool for delivering AD therapy through the BBB and into brain regions where the benefit would be most tangible. However, there is much more work to be done to bring this technology into clinical use [150,151].

## 5. Stem Cells

Safely rejuvenating, rescuing, or replacing the neurons of the brain in AD is the rationale for the use of stem cells [152]. Stem cells can proliferate, self-renew, and differentiate into numerous subtypes characteristic of any of the three germ layers. These properties enable them to serve as suitable reservoirs for cell replacement therapies. Different sources of stem cells with varying capabilities have been identified [153,154]. The primary types of human pluripotent stem cells are (ESCs) and induced pluripotent stem cells (iPSCs) (Figure 2) [155]. Mesenchymal stem cells (MSCs) are multipotent and can transdifferentiate into ectodermal and mesodermal lineages, including neurons [156]. While ESCs are sourced from human embryos, MSCs are taken from adult tissue, while iPSCs represent a conversion of terminally differentiated somatic cells into an ESC-like state. MSCs and iPSCs avoid the ethical problems associated with ESCs [156,157,158,159].

In preclinical studies, ESCs could yield neural progenitor cells (NPCs) when programmed by different growth factors and elements in vitro [160]. In patients with AD, cholinergic neurons in the basal forebrain are lost, and their absence correlates with cognitive decline [161,162]. Bissonnette et al. transformed ESCs into basal forebrain cholinergic neurons and engrafted them onto cultured mouse entorhinal-hippocampal slices ex vivo and showed that these cholinergic neurons promoted functional synapse formation [163]. ESCs were used in vivo in the living mouse brain by Yue et al. [164]. This group produced basal forebrain cholinergic neuron progenitor cells from murine ESCs and transplanted them into the brains of transgenic AD mouse models. These engrafted cells could differentiate into functional cholinergic neurons in the forebrain and improve spatial learning and memory in the mice.

McGinley et al. performed intracranial transplantation of a human neural stem cell line derived from human fetal cortical tissue into an AD mouse model and found that the mice exhibited improved short-term non-associative memory [165,166]. Microscopic examination of the brain showed reduced amyloid burden and increased microglia in the hippocampus and cortex. These benefits were seen even though immunohistochemical studies did not detect the human cells in the murine brain at 17 weeks post-transplant. The authors postulate that even transient exposure to the human ESC cell line was sufficient to confer positive effects.

Neural stem cells extracted from the hippocampus of 1-day old wild type mice were transplanted into the hippocampus of transgenic mice with tauopathy and AD-like traits, including memory impairment, and the mice receiving these stem cells exhibited improvement in short-term memory and decreased accumulation of tau neurofibrillary tangles [167]. A similar study used human ESCs transformed into medial ganglionic eminence (MGE)-progenitor cells, a type of cell that serves as a precursor to basal forebrain neurons. These MGE-like cells were transplanted into a murine model of learning and memory deficits induced by an immunotoxin, which resulted in the correction of memory loss [168].

Although ESCs show potential for treating AD in preclinical studies, their clinical application is limited by ethical issues, risk of teratoma formation, accumulation of mutations, abnormal immune responses, and rejection [169,170]. In addition, despite the advantages of the pluripotent state in ESCs, this property also represents a disadvantage because these cells can undergo genetic alterations leading to tumors or teratomas [171,172]. Therefore, human ESCs as the source of stem cells in treating AD are unlikely. Instead, mesenchymal, and hematopoietic stem cells have been the most widely used and investigated as potential therapeutics for AD [173,174,175].

MSCs are stromal cells derived from various adult sources (blood, adipose tissue, dental pulp) that can differentiate into multi-lineages [176]. These stem cells have a high expansion capacity, low immunogenicity, and low carcinogenic potential [177,178]. With regard to AD pathology in mice, MSCs have been shown to reduce Aβ plaque size, enhance Aβ clearance and reduce Aβ expression [179,180]. MSCs can also alter innate and adaptive immune responses by modulating neuroprotective cytokines such as interleukin (IL)-10 and downregulating pro-inflammatory cytokines such as TNF-α and IL-1β [181]. In addition, human MSCs in culture promote neurogenesis by releasing neurotrophic factors [182]. In preclinical studies, AD mice that received intracerebral transplantation of bone marrow-derived MSCs demonstrated lower Aβ accumulation and increased microglial phagocytic activity [183]. Several preclinical studies have also assessed the efficacy of umbilical cord-derived MSCs obtained from cord lining and Wharton’s Jelly [184]. In mice, human umbilical cord-derived MSCs injected into the carotid artery can migrate into the brain parenchyma. An AD double transgenic mouse model of excessive amyloid deposition injected with these MSCs demonstrated reduced amyloid accumulation, increased microglial activation in the hippocampus and cortex, and better cognitive function during sensorimotor tests compared to AD mouse controls not receiving MSCs [185].

Despite progress in the field of stem cell technology, as demonstrated in preclinical studies using stem cells in animal models of AD, clinical trials assessing the efficacy of this therapeutic remain limited in number. There have been two clinical studies exploring the safety and efficacy of human umbilical cord-derived MSCs in AD patients. The NEUROSTEM-AD treatment, an open-label phase 1 trial (NCT01297218), reported that stereotactic delivery of human umbilical cord-derived MSCs into the hippocampus and precuneus was attainable, safe, and well-tolerated by 9 AD patients [186]. During the first 12-week and last 24-month follow-up periods, no significant adverse effects or dose-limiting toxicity were observed. Results from the trial did show a faster cognitive decline in patients than expected of typical AD progression. Researchers attributed this to the typically faster decline with early onset disease since seven out of the nine enrolled patients had early onset AD.

A second double-blinded, single-centre, open-label phase I/IIa clinical trial (NCT02054208) with 36 months of extended observation (NCT03172117) assessed the safety, dose-related toxicity, and efficacy of human umbilical cord-derived MSCs administered via three intra-cerebro-ventricular (ICV) infusions four weeks apart via an Ommaya reservoir ventricular access device [187,188]. The treatments were given in 2 stages. In the first stage of the study, patients were placed in a low- or high-dose group. In the second stage, patients were randomized into a high-dose or placebo group. Patients developed a transient fever and elevation of cerebrospinal fluid (CSF) white blood cell count after each infusion that resolved rapidly. CSF total tau, p-tau, and Aβ42 were found to be decreased one-day post-infusion but returned to baseline at the 4-week follow-up. This was attributed to the short lifespan of MSCs. A follow-up study will examine neuropsychological scores, imaging, and profiles of biomarkers in these participants compared to the untreated control group.

Human iPSCs, often from fibroblasts, can generate neurons that can be used to study AD processes in human culture systems and cerebral organoids [189,190,191]. There is also the potential for precision medicine studies of unique properties of cells derived from specific patients for evaluation of AD mechanisms [192].

Clinical trials using iPSCs are still rare and not yet being applied in AD, although there are some studies on Parkinson’s disease [193,194,195,196]. Progress in using stem cells in humans is slowed by the disadvantages, such as the need for immunosuppression and the risk of tumor formation with progenitor cells [165]. In addition, the complex anatomy and cellular environment of a patient with AD significantly differ from the homogeneous nature of transgenic animal models developed for the familial type of AD. The precise mechanism and effect of these therapeutics on patients is uncertain.

## 6. Deep Brain Stimulation

Deep Brain Stimulation (DBS) entails electrical brain stimulation using implanted electrodes, subcutaneous leads, and a pulse generator for neuromodulation [197]. This is an invasive modality requiring stereotactic surgical electrode implantation within the brain. The mechanism of action is not well-established, but it has been shown to activate or inhibit brain networks in a way that is postulated to reduce symptoms resulting from circuit issues of the human brain in AD and other disorders such as Parkinson’s disease, essential tremor, primary dystonia, and obsessive-compulsive disorder [198,199,200,201].

In rodent models of AD, DBS has been shown to improve memory, decrease phosphorylated tau and amyloid plaque accumulation and promote cholinergic neurotransmission, hippocampal neurogenesis, and synaptic plasticity [202,203,204]. Within the past ten years, some preliminary clinical trials of DBS in AD demonstrated beneficial effects such as slower cognitive decline, decreased hippocampal atrophy, increased cerebral glucose metabolism and modulation of multi-network brain connectivity in patients suffering from the disease [205,206]. Various stimulation targets of the brain are engaged during DBS treatment in patients with AD. Human clinical trials have used DBS to stimulate the fornix, nucleus basalis of Meynert, and ventral capsule/striatum [205,207,208].

DBS, specifically the fornix, is being investigated as a treatment for patients with mild AD. Results from randomized clinical trials have demonstrated an improvement in cognitive function among some patients and no benefit in others [209]. The fornix, a part of the Papez circuit, is the principal inflow and outflow tract of the hippocampus and middle temporal lobe. Composed of an arcuate fiber bundle that extends from the hippocampus to the mammillary body, the fornix delivers input from the hippocampus to the anterior nucleus of the thalamus. It is responsible for encoding and integrating memory information [210,211]. When this structure is damaged, memory is severely impaired. A transition from mild cognitive impairment to AD is associated with fornix atrophy. Hamani et al. discovered unexpectedly that fornix stimulation could improve memory in a patient who received DBS to treat morbid obesity. Fornix DBS was able to increase recollection and evoke detailed autobiographical memories [212]. Studies in small numbers of subjects have shown that chronic fornix DBS can stabilize or attenuate the rate of memory decline, increase hippocampal volume, and promote cerebral glucose metabolism in AD patients [213,214]. In rodent AD models, chronic fornix DBS improved spatial learning memory and recognition memory, reduced amyloidosis and inflammation and decreased neuronal loss and changes in brain volume [215,216]. Ríos et al. investigated which sites and networks in the brain are the most optimal for DBS in patients with AD. Researchers conducted a post-hoc analysis of data obtained from 46 patients from clinical trials associated with DBS to the fornix (NCT00658125, NCT01608061) [217]. Using structural and functional connectivity data from these trials, the authors reported a strong association with cognitive improvement when stimulated by the Papez and stria terminalis circuits. The most optimal site for stimulation existed at the interface of these two structures.

DBS may have a role in AD treatment, but it cannot be a curative procedure. It can only modulate symptoms. Furthermore, factors in DBS that still need elucidation include stimulation parameters and the exact mechanisms of DBS action in AD [210]. In addition to small sample sizes, a serious limitation of studies conducted thus far is the continued use by participants of acetylcholinesterase inhibitors while receiving DBS therapy. This is confounding because DBS may act, in part, by stimulating the release of acetylcholine [218,219]. DBS is also an invasive procedure with multiple risks, such as bleeding, infection, and other side effects associated with the surgical procedure and the risk of personality changes and depression [220,221,222].

We have now covered the pharmacologic and invasive brain treatments in use or development for AD (summarized in Table 2). In the following sections, we will explore the potential for lifestyle changes to affect cognitive function and their potential to modify AD risk and rate of progression.

## 7. Diet as a Preventative Measure

Measures to delay or prevent the onset of AD have been pursued and tested since the disorder was identified in 1906. Some evidence supports lifestyle adjustments and changes in diet and physical activity level as a viable approach to reducing AD susceptibility [223,224,225]. Epidemiological studies suggest that limiting calories or carbohydrates, raising the intake of certain vitamins and antioxidants, or adjusting the ratio of saturated to unsaturated fats may lower AD risk. However, the true impact of these dietary adjustments is still unresolved, with conflicting data and failure to replicate the preclinical data obtained in animal models [226,227].

### 7.1. Overall Dietary Pattern

The Mediterranean diet and the Dietary Approach to Stop Hypertension (DASH) diet are considered heart-healthy and good for the brain [228,229]. The Mediterranean diet includes vegetables, nuts, seeds, legumes, seafood, olive oil, moderate consumption of dairy and wine, and low meat consumption. The diet contains high omega-3, B vitamins, vitamin D, folic acid, and other necessary nutrients. Low red meat consumption may lead to iron deficiency [230]. The DASH diet is similar but more restrictive in salt, alcohol, and chocolate consumption but allows for more meat. The MIND diet (Mediterranean-DASH Intervention for Neurodegenerative Delay) combines the DASH and Mediterranean diets [231].

Numerous studies show an association between these diets and a lower incidence of AD or MCI with the preservation of cognitive function [232,233,234,235,236,237]. For example, postmortem examination of the brain in persons in the Rush Memory and Aging Project, a long-term study of older adults without dementia at enrollment that includes annual dietary assessments, found that those following the MIND or Mediterranean dietary pattern more rigorously over nearly ten years showed less AD brain pathology and lower amyloid load [238].

Adherence to these plant-forward diets may be especially beneficial when the diet is followed in early adulthood or middle age before cognitive symptoms manifest [239,240,241]. However, some studies show no effect of diet in middle age on dementia and/or AD risk later in life [242].

The DZNE-Longitudinal Cognitive Impairment and Dementia Study (DELCODE), an observational study conducted in Germany, assessed older persons at high risk for AD with extensive neuropsychological testing and a detailed food frequency questionnaire and found that the Mediterranean diet and the MIND diet were associated with better memory and language [243]. Ballarini et al. also used DELCODE data to show a positive association between adherence to a Mediterranean diet and memory performance, and they related these to structural brain images and CSF biomarkers to perform modeling that indicates that this diet may works by preserving brain volume and impacting CSF amyloid and tau biomarkers [244]. Finally, Gregory et al. used data from the European Prevention of Alzheimer’s Dementia Longitudinal Cohort Study (EPAD LCS) to evaluate the effect of the Mediterranean diet on persons living within and outside the Mediterranean region determined to be at risk for AD. They found that following the diet more stringently was associated with better scores on the Four Mountains test, a test of spatial memory, particularly in female participants and within the Mediterranean region [245].

A recent literature review showed an association between lower sodium intake and better cognitive function, but with a modest effect that needs further study and control for confounding variables [246]. In addition, the reports that were evaluated were too heterogeneous for a meta-analysis.

Conversely, a Western type of diet of highly processed foods rich in saturated fats, refined carbohydrates, and salt has been associated with more rapid cognitive decline [247,248,249,250,251]. In addition, the Western diet contributes to obesity and insulin resistance and promotes an inflammatory state, all of which may predispose to the development of AD [252,253,254]. Advanced glycation end products (AGEs) formed in the disrupted metabolic environment of poor glucose control may be one important link between Western diet-induced obesity and cognitive decline [255]. AGEs are present in the tau tangles and amyloid plaques in the AD brain and induce oxidative stress and immune activation in the CNS [256,257,258].

It is essential to recognize that studies involving many foods are especially problematic as different foods within each diet may have a different effect on dementia risk [259].

### 7.2. Calorie Restriction

Calorie restriction has been found to protect against cognitive decline, possibly because it results in decreased systemic inflammation and oxidative stress [260,261,262]. In animal models, calorie restriction is associated with increased longevity, delayed senescence, and neuroprotection [263,264,265]. In addition, it has been shown in humans that restricting calories can improve glucose and lipid metabolism, reduce blood pressure, and decrease biomarkers of inflammation, all of which may support brain health. However, the effects of AD in humans are not proven [266,267,268,269].

### 7.3. Vitamin D

Epidemiological observations have uncovered a neurosteroid hormone vitamin D deficiency in many patients with AD and impaired cognitive function [270,271]. The vitamin D receptor is present in the human brain in neuronal and glial cells, where its activation by vitamin D is important in brain development and function [272,273,274]. A prospective study of 1658 elderly persons without dementia followed for an average of 5.6 years found a substantial increase in the risk of developing AD and all-cause dementia with vitamin D deficiency [275]. Meng et al. performed a two-sample randomization analysis looking at associations between vitamins and AD and found low vitamin D levels causally associated with increased AD risk [276]. Multiple meta-analyses have also shown a link between low circulating levels of vitamin D and AD [277,278,279]. The association is particularly strong when vitamin D deficiency is profound, with levels below 10 ng/mL [280,281]. However, other studies have failed to find a clear benefit in AD risk reduction with vitamin D supplementation in older adults [282,283,284].

Several neuron-preserving effects of vitamin D have been shown in animal models, and these support the importance of achieving sufficient serum levels of this compound. Among these neuroprotective properties is the ability of vitamin D to reduce inflammation and oxidative stress and to regulate calcium homeostasis [285,286,287,288].

In murine models, vitamin D reduces Aβ plaque build-up and promotes degradation [289,290,291]. Furthermore, the prevention of Aβ accumulation is attributed to augmented expression levels of APP and BACE1 by vitamin D [292].

### 7.4. The B Vitamins: B6 (Pyridoxine), Folate (B9), B12 (Cobalamin)

The roles of folate, vitamin B6, and vitamin B12 have been scrutinized because these vitamins have links to CNS function, and deficiencies are common in older persons [293,294]. A de Wilde et al. meta-analysis found that vitamin B12 and folate availability in the brain and circulation is lower in AD patients than in controls [295].

These vitamins participate in the linked cycles of folate and methionine metabolic pathways with the consumption of homocysteine, a key step accomplished by cyclative methylation of homocysteine to methionine. In insufficient B6, B12 and/or folate, hyperhomocysteinemia occurs and may be associated with cognitive impairment in later life [296,297,298,299,300]. However, the efficacy of these vitamins in reducing elevated homocysteine and preventing or slowing AD progression is unclear. Results of multiple studies of AD and MCI patients supplemented with these B vitamins have been conflicting. Many have failed to demonstrate slowing of cognitive decline [301,302].

On the other hand, a randomized study of 240 MCI patients found that folate and vitamin B12 in combination reduced inflammatory markers and improved cognition after six months [303]. Another recent study of 120 AD patients, half randomized to receive B12, and folate and the other half randomized to receive a placebo over six months, found that supplementation with these vitamins improved cognitive performance [304]. However, these patients were not on a folate-fortified diet before enrollment, which may have allowed the needed contrast with newly added folate.

Sufficient levels of vitamin B6 are essential for CNS function because this vitamin is a coenzyme in numerous reactions involving amino acid production, a required cofactor for the synthesis of dopamine, and it plays a crucial role in the synthesis of γ-aminobutyric acid (GABA), the main CNS inhibitory neurotransmitter [305]. Vitamin B6 may thus counteract nerve damage by limiting excitotoxicity [306]. In addition, folate is essential in modulating homocysteine levels, and it reduces oxidative stress, but its ability to lower inflammatory cytokine levels is in dispute [307,308].

Vitamin B12 plays a role in the cellular metabolism of carbohydrates, proteins and lipids, and its deficiency has neurologic consequences that can include cognitive decline [309,310,311]. In addition, vitamin B12 has anti-oxidant properties postulated to be neuroprotective [312]. Politis et al. found an association between low serum B12 and higher peripheral blood mononuclear cell production of Il-6, an inflammatory cytokine [313]. Song et al. showed that high homocysteine and low B12 levels were linked to temporal lobe atrophy in AD subjects [314]. A case-control study from Shrestha and colleagues with a sample size of 90 found a significant association between vitamin B12 deficiency and AD after adjusting for age [315].

More research is required to determine whether the association between the B vitamins and cognition indicates a path to treatment. The studies thus far point to the importance of maintaining the level of these vitamins in the normal range in older persons and to the cooperative nature of their activity.

### 7.5. Antioxidants

An imbalance between the production of reactive oxygen species and the ability of the brain to generate an anti-oxidant defence is widely thought to contribute to AD pathophysiology [316,317]. In addition, oxidative stress can damage neurons through disruption of the mitochondrial respiratory chain, protein and lipid peroxidation, and induction of neuronal apoptosis [318,319]. Based on these accumulated findings, anti-oxidative stress therapy could be beneficial in preserving neurons in AD. However, this data is mixed, and the issue is unresolved [320,321]. Beydoun et al. used the Third National Health and Nutrition Examination Survey (NHANES III) to examine interactions between serum nutritional biomarkers of antioxidant status in relation to AD in a selection of adults over 45. Although incident all-cause dementia was inversely associated with serum lutein + zeaxanthin and β-cryptoxanthin levels, no significance was found with AD-specific dementia [322]. However, they did find an antagonistic interaction between vitamin E and lycopene in relation to AD incidence. Another study utilized The Healthy Aging in Neighborhoods of Diversity across the Life Span (HANDLS) study to examine diet and cognition longitudinally and found a link between vitamin E consumption and greater verbal memory performance [323]. In a multi-centre clinical trial that randomly assigned 78 AD subjects to 16 weeks of treatment with either vitamin E + vitamin C + α-lipoic acid or Coenzyme Q or placebo, results were not encouraging. Antioxidants did not improve CSF amyloid or tau biomarkers, and the cognitive decline accelerated in the vitamin E + vitamin C + α-lipoic acid group.

## 8. Mental and Physical Activity

### 8.1. Exercise and Physical Activity

Multiple studies have repeatedly demonstrated that increased physical and mental activity is associated with a decreased risk of AD [324].

Exercise and diet may forestall AD symptoms [225,325,326,327]. Exercise can attenuate some known AD risk factors, including hypertension, hyperglycemia, and obesity [328,329]. Exercise can also improve cerebral blood flow [330]. It is estimated that non-demented persons who engage in regular physical activity reduce their risk of cognitive decline by more than 25% compared to sedentary persons, and effects exceed 30% when the activity level is high [331,332]. In addition, physical activity may help to preserve executive function in persons with dementia [333]. Walking alone was recently shown in a pilot study to improve cognitive performance in a small sample of MCI patients [334].

Exercise can prevent or delay the loss of brain volume and improve the functional connectivity of brain regions [335,336]. In addition, exercise may reduce oxidative stress. However, studies in humans have not found exercise to consistently improve levels of BDNF, a neurotropic factor important in maintaining synaptic function and neuronal plasticity [337,338,339,340,341].

People over age 65 are often increasingly sedentary [342]. Numerous studies have indicated that certain measures of gait can predict future cognitive and functional decline [343]. Furthermore, cross-sectional, and longitudinal studies have associated gait abnormalities with imaging, biofluid, and genetic markers of AD across all stages [343]. Exercise for older persons may be difficult due to functional limitations, painful joints, fear of falling and other issues [344]. Considering these issues is important in removing barriers to optimize participation in physical activity by older adults [345].

### 8.2. Mental Exercise

Researchers have also questioned whether cognition-focused interventions can lower the risk of AD or at least help to maintain cognitive reserve [346,347]. Higher education level, which may covary with regular mental exercise, has also been associated with a reduced risk of dementia [348]. Many physicians recommend that individuals of all ages perform word searches, sudoku, crossword puzzles, and other word-matching games. In addition, computer programs and virtual reality experiences are designed to challenge the brain [349,350]. The benefits of mental exercise to the AD brain are uncertain, but cognitive stimulation may be helpful, particularly in MCI patients [351,352,353,354]. Studies are underway or planning to evaluate the combination of mental and physical challenges using virtual reality in persons with mild AD [355,356].

In summary, lifestyle adjustments may have value in delaying AD onset (Table 3). For example, maintaining overall good health by incorporating physical and mental activity combined with a nutritious diet can provide the brain with a nourishing and sustaining environment but is limited in how much it can alter the course of AD.

## 9. The Future

Unraveling the intricacies of AD etiopathogenesis is an arduous but not insurmountable task that has been approached in multiple ways, as illustrated in this review. However, to find the breakthrough that is so urgently needed, the evidence supports a move away from simplistic attempts to lower amyloid or tau production and perhaps to move on to a more complex strategy that preserves neuron longevity, modulates autophagy, and maintains mitochondrial integrity and bioenergetic functions [357,358].

Valuable clues can be garnered from families with inherited forms of AD. There are ways that the human genetic makeup can forestall AD symptoms in the face of familial AD. Persons carrying a mutation in the presenilin one gene that causes a substitution of the 280 Glutamic acids by Alanine (*E280A*) in the encoded protein exhibit an autosomal dominant form of early onset AD with complete penetrance by the time the patients reach their early seventies in age [359,360]. In those harboring this mutation, the onset of dementia is delayed for those who also carry specific apoE alleles, including the apoe2 allele and the apoE3 Christchurch mutation [361,362]. Lopera et al. showed that heterozygosity for a rare variant (H3447R) in the gene for reelin, an extracellular matrix protein and a ligand that binds apoE, also delays AD symptoms in a person carrying the *E280A* mutation [363]. Insights such as these give hope that a mechanical model of AD can be built, and with a better understanding, real headway can be made. In addition to natural mutations in humans, we can also learn from AD brain models constructed in cell culture that may mimic many properties of the human brain [364].

## 10. Conclusions

The incidence of AD has steadily increased in the past few decades, affecting up to 50% of people 85 years of age and older. Current therapies include acetylcholinesterase inhibitors, N-methyl d-aspartate receptor antagonists, and, more recently, anti-amyloid antibodies. However, the effectiveness of these therapeutic strategies is limited, none are curative, and they are variably palliative. This paper analyzes more novel potential strategies beyond the attenuation of amyloid and tau accumulation. Novel anti-neuroinflammatory drugs and repurposing of currently available anti-inflammatory drugs, such as TNF-α inhibitors, are just some strategies discussed in this paper. The potential effects on the brain of systemic processes involving glucose metabolisms and energy production, such as T2DM and metabolic syndrome, are explored, and the possible role of the microbiome-gut-brain axis in the pathogenesis of AD is covered. The effect of deficiencies in organic compounds and the role of modifiable factors like diet and exercise in the progression of cognitive decline are considered. Strategies aimed at safely replacing affected neurons via stem cells and effectively delivering these therapeutics via lipophilic and biocompatible nanoparticles are also discussed; Although preclinical animal work involving stem cell transplantation shows promise, clinical testing is the next step. The pressing need for effective medical treatment requires further research and a better understanding of the fundamental mechanisms involved in the AD process. Extensive effort and determination are essential in the search for a significant breakthrough.

## Figures and Tables

**Figure 1 medicina-59-01084-f001:**
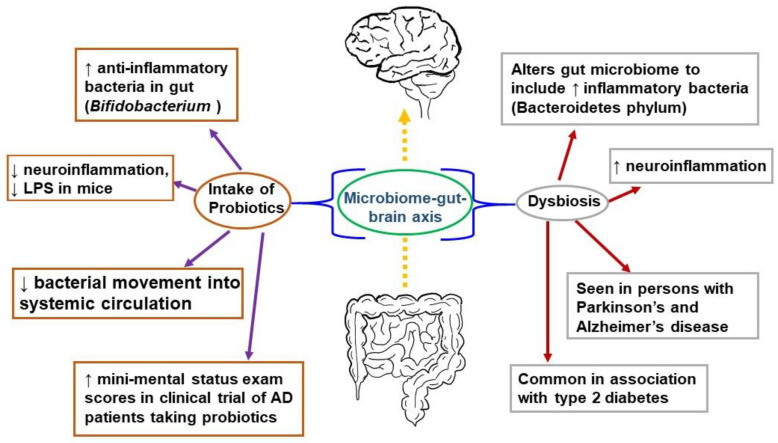
The microbiome-gut-brain axis is a potential pathological mechanism in AD. The gut microbiome comprises numerous bacterial species in a symbiotic relationship with the human organism. It helps protect the host from bacterial overgrowth and carcinogens via the secretion of short-chain fatty acid metabolites. Dysbiosis occurs when the gut microbiome is negatively altered and exhibits reduced species diversity. This, in turn, can promote the development of metabolic syndrome, the growth of inflammatory bacteria, and neuroinflammation. To combat dysbiosis, probiotics can support the growth of anti-inflammatory bacteria, decrease neuroinflammation, and improve mini-mental status scores among patients with AD.

**Figure 2 medicina-59-01084-f002:**
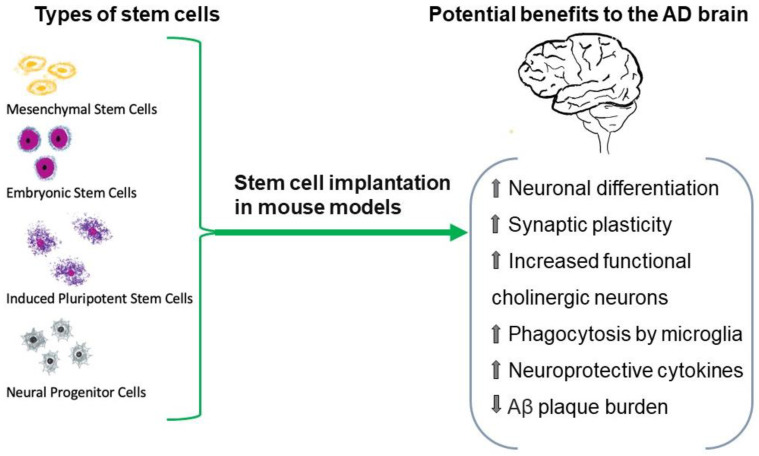
Stem cells are being explored as an avenue of AD treatment. Several sources of these pluripotent cells have been identified. Preclinical studies suggest that stem cells may be able to rejuvenate, rescue, and replace unhealthy neurons. In addition, transplantation of these cells into specific brain regions may yield benefits, as shown in this figure. However, more human clinical trials are needed for definitive answers.

**Table 1 medicina-59-01084-t001:** Potential Therapeutics for the Management of Neuroinflammation in AD.

Targets	Drugs	Modulation of Neuroinflammation
COX-1 and COX-2 inhibitors	NSAIDs (diclofenac/misoprostol, nimesulide, naproxen, rofecoxib, ibuprofen, indomethacin, tarenflurbil, and celecoxib)	COX-2 overexpression is seen in activated microglia. Potential COX-2 inhibition might reduce neuroinflammatory mediators and prostaglandin release by these cells.
TNF-α inhibitors	Etanercept, infliximab, XPro1595	Activated microglia promote the TNF-α and TNF receptor 1 axis to induce a neuroinflammatory state.
TREM2 agonists	(AL002a)—TREM2 mouse IgG1 antibody agonist (AL002c)—mouse IgG1 anti-human TREM2 monoclonal antibody agonist	Genetic mutations in TREM2 receptors are associated with AD. Activation of TREM2 is neuroprotective.
CD33 inhibitors	AL003—antibody against CD33 receptor	Higher CD33 levels and subsequent activation of CD33+ microglia are associated with higher Aβ plaque burden.
Filamin A conformation restoration	PTI-125—a small molecule drug that interacts with Filamin A to reestablish its native state	Altered filamin A promotes the hyperphosphorylation of tau by activating the signaling of Aβ42 using the α7-nicotinic acetylcholine receptor

Abbreviations: COX—cyclooxygenase; NSAIDs—non-steroidal anti-inflammatory drugs; TNF—tumor necrosis factor; TREM2—Triggering Receptor Expressed on Myeloid Cells 2; IgG1—immunoglobulin G1; Aβ—amyloid β.

**Table 2 medicina-59-01084-t002:** Experimental treatment approaches for Alzheimer’s disease.

Category of Method	Specific Intervention	FDA Approved	Clinical Utility or Value	Side Effects	Potentially Disease-Modifying	References
Anti-amyloid	Aducanumab, lecanemab	Accelerated approval	Limited	Infusion reaction, headache, ARIA, brain swelling, brain hemorrhage	Yes	[9,10,11,12,13]
Treat CNS insulin resistance	Insulin, metformin	No	Unproven	Hypoglycemia with insulin, GI effects of metformin	Yes	[108,109,110,111,112,113,114,115,116]
Stem cells	ESCs, MSCs, iPSCs	No	Unproven	Risks from immunosuppression, tumor formation with ESCs, infection, bleeding	Yes	[161,183,186,187,188,192,193,194]
Deep brain stimulation	Delivery of electrical pulses to a defined area of the brain	No	No	Requires implant of the electrode, headache, infection, brain hemorrhage	No	[197,198,199,200,201,202,203,204,205,206,220,221,222]

CNS: the central nervous system; GI: gastrointestinal; ESCs: embryonic stem cells; MSCs: mesenchymal stem cells; iPSCs: induced pluripotent stem cells; ARIA: amyloid-related imaging abnormalities.

**Table 3 medicina-59-01084-t003:** Lifestyle modifications for prevention and treatment of Alzheimer’s disease.

Lifestyle Change	Specific Intervention	Clinical Utility or Value	Side Effects	Potentially Disease-Modifying	References
Alter gut microbiome	Consumption of probiotics and prebiotics. Fecal transplant.	Unproven	Gas, bloating, constipation, nausea, allergic reactions.	Yes	[130,138,139]
Change overall diet	Mediterranean diet, DASH diet, MIND diet	It may preserve memory and lower dementia risk	A Mediterranean diet low in iron	Yes	[230,231,232,233,234,235,236,237,239,240,241]
Calorie restriction	Intermittent fasting	Unproven	Hunger, nutritional deficiencies	Maybe	[260,261,262,266,267,268,269]
Physical activity, exercise	Structured activity program, non-sedentary lifestyle	May preserve executive function	Risks from falls	Yes	[325,326,327,331,332,333,344]
Mental	Cognitive challenges, puzzles, memory tasks, matching tasks, and spatial recognition tasks.	Unproven	None	Maybe	[346,347,348]

CNS: the central nervous system; GI: gastrointestinal; ESCs: embryonic stem cells; MSCs: mesenchymal stem cells; iPSCs: induced pluripotent stem cells; ARIA: amyloid-related imaging abnormalities; DASH: Dietary Approach to Stop Hypertension; MIND: Mediterranean-DASH Intervention for Neurodegenerative Delay.

## Data Availability

Not applicable.

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
