# Peer review of "Alzheimer’s Disease Treatment: The Search for a Breakthrough"

_medicina, 2023, doi:10.3390/medicina59061084_

Round 1
Reviewer 1 Report
The research under review provides a broad overview of various pre-clinical treatments for Alzheimer's disease, including the inhibition of microglial receptors, modulation of the microbiome-brain-gut axis, and increased exercise, both physical and mental. These interventions aim to optimize brain health by attenuating inflammation and enhancing toxin-clearing autophagy. While the review offers some valuable insights, there are several areas that could benefit from improvement.
Firstly, it is not clear what the main innovation or contribution of this review is, and how it stands out compared to other reviews in this area. While the paper provides a comprehensive summary of current pre-clinical treatments for Alzheimer's disease, it is unclear what new knowledge or insights the review offers beyond what has already been published.
Secondly, the review would benefit from a more systematic comparison of the different treatments discussed. While the paper provides some comments on each intervention separately, there is little analysis of the relative strengths and limitations of each approach. A more detailed comparison of the different interventions would provide a more nuanced understanding of their potential efficacy and identify areas for future research.
In summary, while the research under review provides a useful overview of current pre-clinical treatments for Alzheimer's disease, there is room for improvement.
Author Response
RE: Review Article Manuscript ID: medicina-2392593 “Alzheimer’s Disease Treatment: The Search for a Breakthrough”
We thank the reviewer for thoroughly scrutinizing our manuscript. As requested, we have revised the manuscript and addressed the specific comments of the reviewer. The revised sections are delineated in red in a marked copy of the manuscript text.
Below, we provide a point-by-point response to the reviewer’s comments.
Reviewer # 1 Comments
- COMMENT #1: Firstly, it is not clear what the main innovation or contribution of this review is, and how it stands out compared to other reviews in this area. While the paper provides a comprehensive summary of current pre-clinical treatments for Alzheimer's disease, it is unclear what new knowledge or insights the review offers beyond what has already been published.
RESPONSE: This review is designed to present a comprehensive overview of the current state of knowledge as an anchor for the special issue "Commemorative Issue Celebrating the 20th Anniversary of the Alzheimer’s Foundation of America: Understanding and Treating Alzheimer’s Disease". Two of the authors (Reiss and Pinkhasov) are editors of the Special Issue. This is a paper that we believe will be of interest to readers of Medicina at this time when the field of Alzheimer’s disease is at a crossroads. We have added several summarizing paragraphs as well as a section on “The Future” that we hope will tie things together. Our emphasis on tangible progress in humans gives us a perspective not always found in this type of review.
- COMMENT #2: Secondly, the review would benefit from a more systematic comparison of the different treatments discussed. While the paper provides some comments on each intervention separately, there is little analysis of the relative strengths and limitations of each approach. A more detailed comparison of the different interventions would provide a more nuanced understanding of their potential efficacy and identify areas for future research.
RESPONSE: Thank you for this excellent suggestion. We have added 2 new tables (Table 2 and Table 3) incorporating this type of comparison.

Reviewer 2 Report
I must congratulate the authors for their work. It is an interesting manuscript about the frontiers of Alzheimer’s Disease management. I believe that the authors answered their main query. The length of the manuscript, as well as the references, are appropriate.
1. What is the main question addressed by the research?
The authors provide new insights into possible targets for AD management. It is an extensive review of the different pathophysiological explanations.
2. Do you consider the topic original or relevant in the field? Does it
address a specific gap in the field?
Yes, it is an original and relevant topic addressing a specific field gap. It is possible that the authors reviewed the main gaps for a future original study.
3. What does it add to the subject area compared with other published material?
The manuscript describes the most extensive idea of therapeutical management of AD. It is a fundamental article including different hypothetical mechanisms.
4. What specific improvements should the authors consider regarding the methodology? What further controls should be considered?
In my opinion, the manuscript is appropriate in its present form.
5. Are the conclusions consistent with the evidence and arguments presented and do they address the main question posed?
Yes, the conclusions are consistent, and the references provided are appropriate. I performed an analysis for auto-citation, and the number was less than 3%.
6. Are the references appropriate?
Yes, the answer is above.
7. Please include any additional comments on the tables and figures.
I believe that no additional comment would improve the quality of the manuscript. Maybe the authors provide a video explaining the different hypotheses. But, in the present form, the manuscript is appropriate, including language and description.
Author Response
RE: Review Article Manuscript ID: medicina-2392593 “Alzheimer’s Disease Treatment: The Search for a Breakthrough”
We thank the reviewer for thoroughly scrutinizing our manuscript. As requested, we have revised the manuscript and addressed the specific comments of reviewer #1. The revised sections are delineated in red in a marked copy of the manuscript text.
Reviewer # 2 Comments
- COMMENT: The authors provide new insights into possible targets for AD management. It is an extensive review of the different pathophysiological explanations. I believe that no additional comment would improve the quality of the manuscript. Maybe the authors provide a video explaining the different hypotheses. But, in the present form, the manuscript is appropriate, including language and description.
RESPONSE: We thank the reviewer for the favorable comments and for the understanding of our goals in writing this manuscript.

Round 2
Reviewer 1 Report
The authors have addressed my previous concerns and made valuable additions to the paper. The response provided clarifications regarding the main innovation and contribution of the review, emphasizing its relevance to the special issue on Alzheimer's disease. The inclusion of summarizing paragraphs and a section on "The Future" helps tie the information together effectively.